# Virulence and antimicrobial resistance profile of non-typhoidal *Salmonella enterica* serovars recovered from poultry processing environments at wet markets in Dhaka, Bangladesh

**Nure Alam Siddiky**[1], **Samun Sarker**[1], **Shahidur Rahman Khan**[2], **Tanvir Rahman**[2], **Abdul Kafi**[2], **Mohammed A. Samad**[1]*

**1** Antimicrobial Resistance Action Center, Bangladesh Livestock Research Institute, Savar, Dhaka, Bangladesh, **2** Department of Microbiology and Hygiene, Bangladesh Agricultural University, Mymensingh, Bangladesh

* msamad@blri.gov.bd

**Data Availability Statement:** All relevant data are within the paper and its Supporting information files.

## Abstract

The rapid emergence of virulent and multidrug-resistant (MDR) non-typhoidal *Salmonella* (NTS) *enterica* serovars is a growing public health concern globally. The present study focused on the assessment of the pathogenicity and antimicrobial resistance (AMR) profiling of NTS *enterica* serovars isolated from the chicken processing environments at wet markets in Dhaka, Bangladesh. A total of 870 samples consisting of carcass dressing water (CDW), chopping board swabs (CBS), and knife swabs (KS) were collected from 29 wet markets. The prevalence of *Salmonella* was found to be 20% in CDW, 19.31% in CBS, and 17.58% in KS, respectively. Meanwhile, the MDR *Salmonella* was found to be 72.41%, 73.21%, and 68.62% in CDW, CBS, and KS, respectively. All isolates were screened by polymerase chain reaction (PCR) for eight virulence genes, namely *inv*A, *agf*A, *lpf*A, *hil*A, *siv*H, *sef*A, *sop*E, and *spv*C. The *S.* Enteritidis and untyped *Salmonella* isolates harbored all virulence genes while *S.* Typhimurium isolates carried six virulence genes, except *sef*A and *spv*C. Phenotypic resistance revealed decreased susceptibility to ciprofloxacin, streptomycin, ampicillin, tetracycline, gentamicin, sulfamethoxazole-trimethoprim, amoxicillin-clavulanic acid, and azithromycin. Genotypic resistance showed a higher prevalence of plasmid-mediated *bla*TEM followed by *tet*A, *sul*1, *sul*2, *sul*3, and *str*A/B genes. The phenotypic and genotypic resistance profiles of the isolates showed a harmonic and symmetrical trend. According to the findings, MDR and virulent NTS *enterica* serovars predominate in wet market conditions and can easily enter the human food chain. The chi-square analysis showed significantly higher associations among the phenotypic resistance, genotypic resistance and virulence genes in CDW, CBS, and KS respectively ($p < 0.05$).

**Funding:** The authors received no specific funding for this work.

**Competing interests:** The authors have declared that no competing interests exist.

## Introduction

*Salmonella* has been recognized as one of the common pathogens that cause gastroenteritis [1, 2] with significant morbidity, mortality, and economic loss [3, 4]. WHO reported 153 million cases of NTS enteric infections worldwide in 2010, of which 56,969 were dead along with 50% were foodborne [5]. The disease surveillance report of China from 2006 to 2010 identified *Salmonella* as the second foodborne outbreak [6]. NTS serovars like Typhimurium and Enteritidis are the predominant worldwide among the 2,600 serotypes of *Salmonella* that have been identified [7, 8]. Poultry has been regarded as the single prime cause of human salmonellosis and avian salmonellosis is not only affects the poultry industry but also can infect humans and caused by the consumption of contaminated poultry meat and eggs [9]. The eggs are considered to be the primary cause of salmonellosis and numerous other foodborne outbreaks [10–13]. Generally, *Salmonella* grows in animal farms may contaminate eggs and/or meat during the slaughtering process before being transferred to humans through the food chain. Indeed, numerous previous studies have been reported the isolation of *Salmonella* from foods of animal origin as well as human samples [14–17]. Human *S.* Enteritidis are generally linked with the consumption of contaminated eggs and poultry meat, while *S.* Typhimurium with the consumption of pork, poultry, and beef [18, 19]. Different prevalence of *Salmonella enterica* serovars has been reported around the globe from animal products and by-products [18, 20, 21]. *Salmonella* Typhimurium and Enteritidis are the most frequently reported serovars associated with human foodborne illnesses [22]. Untyped *Salmonella* of animal origin has been increasingly observed in Bangladesh [23, 24] but limited information has been published on *Salmonella enterica* serovars isolated from chicken processing environments.

Widespread uses of antimicrobials in poultry farming generate benefits for producers but aggravate the emergence of AMR bacteria [25]. Microorganisms that develop AMR are sometimes referred to as superbugs and open the door to treatment failure for even the most common pathogens, raise health care costs, and increases the severity and duration of infections. AMR burden may kill 300 million people during the next 35 years with a terrible impact on the global economy declining GDP by 2–3% in 2050 [26]. WHO recognized AMR as a serious threat, is no longer a forecast for the future, which is happening around the world and affects everybody regardless of age, sex, and nation [27]. Misuse and overuse of existing antimicrobials in humans, animals, and plants are accelerating the development and spread of AMR [28]. Antimicrobials are used in Bangladesh as the therapeutic, preventive, and growth promoters in the poultry production system [29]. The problem of AMR *Salmonella* emerged global concern in the modern decade [24, 30]. MDR *Salmonella* of poultry origin has been increasing in Bangladesh [31, 32].

Usually, the virulence factors promote the pathogenicity of *Salmonella* infection. Chromosomal and plasmid-mediated virulence factors are associated with the pathogenicity of *Salmonella*. *Salmonella* possesses major virulence genes such as *inv*A, *agf*A, *Ipf*A, *hil*A, *siv*H, *sef*A, and *sop*E. The infectivity of *Salmonella* strains is related with different virulence genes existent in the chromosomal *Salmonella* pathogenicity islands (SPIs) [33]. The attack qualities invA, hilA, and sivH code with a protein within the inward chromosomal membrane of Salmonella that's essential for the intrusion to epithelial cells [34]. Moreover, *Salmonella* effector protein attached by *sop*E gene which have potential to *Salmonella* virulence [35]. The plasmid-mediated *spv*C gene is liable for vertical transmission of *Salmonella* [36]. The long polar fimbria (Ipf operon) make the fascination of the organisms for Peyer's patches and attachment to intestinal M cells [37]. The aggregative fimbria (agf operon) advances the essential interaction of the *Salmonella* with the digestive system of the host and invigorate microbial self-aggregation for higher rates of survival [38]. The Salmonella-encoded fimbria (sef operon) supports

interaction between the organisms and the macrophages [38]. In spite of the fact that was a paucity of information in the determination of virulence gene from *Salmonella enterica* serovars in Bangladesh but recently eight virulence genes were found in *Salmonella* isolates of poultry origin in Bangladesh [32].

Wet markets are very common in Bangladesh which are commonly dirty, chaotic, and unhygienic and floors are constantly sprayed with water for washing and to conserve the humidity [39]. Dressing and processing of poultry in the open environment are common practices in the traditional wet markets. The chicken vendors himself dressing the chicken without having personal protective devices, without using clean dressing utensils such as chopping boards and knives. Even the same water is used frequently for washing or cleaning the whole dressed carcass. There is a great possibility of cross-contamination and horizontal distribution of MDR Salmonella in the environment of wet markets [40]. The whole chicken carcass, vendor, and the consumer may be infected with *Salmonella* due to poor sanitary and hygienic practices. Even there is a great scope to spread and transmission of *Salmonella enterica* serovars in the agricultural food chain in the wet markets since most of the products are sold at room temperature and exposed to the environment [41]. A previous study stated that the incidence of *Salmonella* at different sites of wet markets has indicated a cause of cross-contamination in the meat during sale through food or equipment contact surfaces [39]. Based on the importance of foodborne *Salmonella* at wet markets, this study aimed at determining the pathogenicity and profile of antimicrobial susceptibility of *Salmonella* enterica serovars isolated from poultry processing environments in the wet markets of Dhaka, Bangladesh.

## Materials and methods

### Study design and sample collection

The study was conducted in the 29 chicken wet markets around Dhaka city, the capital of Bangladesh from February to December 2019 in a cross-section manner (Fig 1). Dhaka city is called the biggest chicken selling hub due to the mass population density and economic sovereignty of the population. The sample size was calculated by using the "sample size calculator for prevalence studies, version 1.0.01" based on the 25% prevalence of *Salmonella* spp. reported previously in Bangladesh [42, 43]. The desired individual sample number should not be less than 289. Three types of poultry processing environmental samples consisting of carcass dressing water (CDW), chopping board swabs (CBS), and knife swabs (KS) were collected independently as the number of 290. The ten samples of each three types (CDW, CBS and KS) were collected from each site on a single visit. The sterile cotton swabs contained in 10 ml buffered peptone water (BPW) were used for swabbing the samples. The samples were collected aseptically and immediately brought to the Antimicrobial Resistance Action Centre (ARAC) with an insulated icebox. This study received ethical approval from the Ethical Committee of the Animal Health Research Division at the Bangladesh Livestock Research Institute (BLRI), Dhaka, Bangladesh (ARAC: 15/10/2019:05).

### *Salmonella* isolation and identification

*Salmonella* isolation and identification was carried out according to the guidelines of ISO [44] as follows; pre-enrichment of the swab smear in BPW (Oxoid, UK) followed by aerobic incubation at 37°C for 18–24 h. Further, 0.1 mL of the pre-enriched sample was positioned discretely into three different locations on Modified Semisolid Rappaport Vassiliadis (MSRV; Oxoid, UK) agar and incubated at 41.5°C for 20–24 h. Further, a single loop of MSRV cultured medium was taken and subsequently smeared onto Xylose Lysine Deoxycholate (XLD; Oxoid, UK) and MacConkey agar (Oxoid, UK) medium and overnight incubated at 37°C. The typical

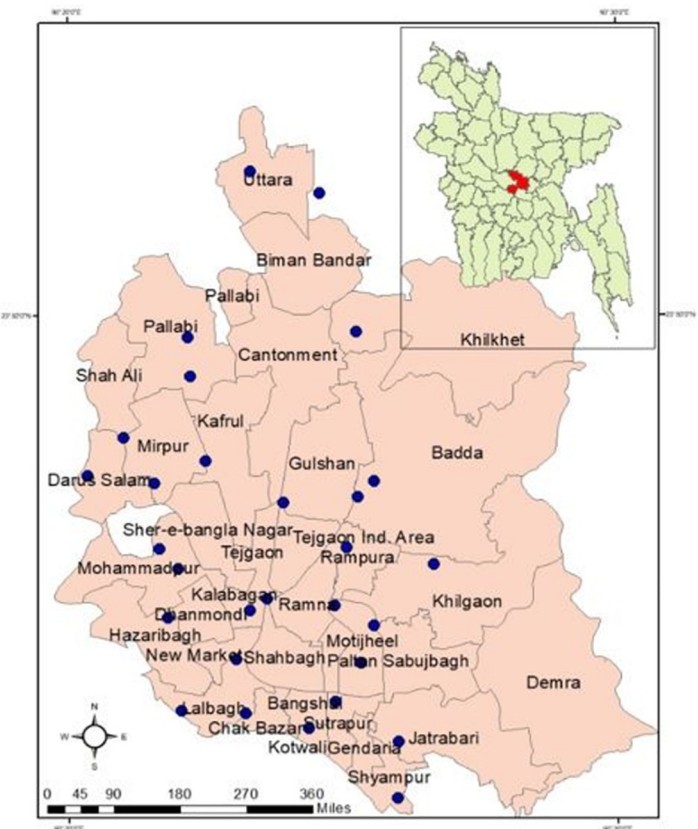

**Fig 1. Map of the Dhaka city with study locations (blue colored circles) of 29 wet markets.**

black centered colony with a reddish zone on XLD and a colorless colony on MacConkey were extracted and subsequently sub cultured in nutrient agar (NA; Oxoid, UK) medium. The biochemical conformation was done by triple sugar iron (TSI), motility indole urea (MIU), catalase and oxidase tests. Final confirmation was done by the mechanical Vitek-2 compact analyzer (bioMérieux, France) as well as molecular detection by the polymerase chain reaction (PCR) method [32].

## DNA extraction

The conventional boiling method was used for the extraction of DNA followed by a proven procedure as applied earlier [32, 45, 46]. Concisely, the pure *Salmonella* isolate was cultured on nutrient agar medium and subsequently overnight incubated at 37°C. A few fresh and juvenile colonies were harvested from overnight culture and suspended in nuclease-free water. Then the bacterial suspension was boiled at 99°C for 15 min followed by chilled on ice for a short duration. Lastly, the debris was separated by high speed centrifugation and the supernatant was taken as the DNA template for further PCR assay.

## PCR detection of *Salmonella* and *Salmonella enterica* serovars

Uniplex PCR (U-1) was performed to detect *Salmonella* species targeting virulence gene *inv*A [47]. Multiplex PCR (M-I) was done to detect *S.* Typhimurium and *S.* Enteritidis [48, 49]. PCR

reaction was adjusted in 25 μL mixture containing 2 μL of DNA template, 12.5 μL of 2x master mix (Go Taq Green Master Mix, Promega), 0.5 μL each of forward and reverse primers (10 pmol/μL) and 9.5 μL nuclease-free water. The PCR products were run at 100 V with 500 mA for 30 min in 1.5% agarose gel containing ethidium bromide. A 100bp DNA ladder (Thermo Scientific, USA) was used as a size marker. The primers used to detect *Salmonella* and *Salmonella enterica* serovars are presented in Table 1. The ATCC of *S*. Typhimurium (ATCC-14028) and *S*. Enteritidis (ATCC-13076) were used as a positive control. Consequently, PCR positive *Salmonella* serovars Typhimurium and Enteritidis was further reconfirmed by the Vitek-2 compact analyzer (bioMérieux, France).

**Table 1. Primers used to detect *Salmonella enterica* serovars and resistance genes.**

| PCR | Target gene | Sequence | Amplicon size (bp) | Thermal Profile | References |
|---|---|---|---|---|---|
| **U-1** | *inv*A | F-GTGAAATTATCGCCACGTTCGGGCAA | 284 | 95°C for 1 min; 38 cycles of 95°C for 30s, 64°C for 30s and 72°C for 30s; elongation step at 72°C for 4 min | [58] |
| | | R-TCATCGCACCGTCAAAGGAACC | | | |
| M-1 | Typh | F-TTGTTCACTTTTTACCCCTGAA | 401 | 95°C for 2 min; 30 cycles of 95°C for 1 min, 57°C for 1 min and 72°C for 2 min; elongation at 72°C for 5 min | [49] |
| | | R-CCCTGACAGCCG TTAGATATT | | | |
| | *sdf*-1 | F-TGTGTTTTATCTGATGCAAGAGG | 293 | | [48] |
| | | R-CGTTCTTCTGGTACTTACGATGAC | | | |
| M-II | *bla*TEM | F-CCGTTCATCCATAGTTGCCTGAC | 800 | 94°C for 10 min; 30 cycles of 94°C for 40 s, 60°C for 40 s and 72°C for 1 min; elongation step at 72°C for 7 min | [56] |
| | | R-TTTCCGTGTCGCCCTTATTC | | | |
| | *bla*SHV | F-AGCCGCTTGAGCAAATTAAAC | 713 | | [56] |
| | | R-ATCCCGCAGATAAATCACCAC | | | |
| | *bla*OXA | F-GGCACCAGATTCAACTTTCAAG | 564 | | [56] |
| | | R-GACCCCAAGTTTCCTGTAAGTG | | | |
| M-III | *bla*CTX-M-1 | F-TTAGGAARTGTGCCGCTGYA | 688 | 94°C for 10 min; 30 cycles of 94°C for 40 s, 60°C for 40 s and 72°C for 1 min; elongation step at 72°C for 7 min | [56] |
| | | R-CGATATCGTTGGTGGTRCCAT | | | |
| | *bla*CTX-M-2 | F-CGTTAACGGCACGATGAC | 404 | | [56] |
| | | R-CGATATCGTTGGTGGTRCCAT | | | |
| | *bla*CTX-M-9 | F-TCAAGCCTGCCGATCTGGT | 561 | | [56] |
| | | R-TGATTCTCGCCGCTGAAG | | | |
| | *bla*CTX-Mg8/ 25 | F-AACRCRCAGACGCTCTAC | 326 | | [56] |
| | | R-TCGAGCCGGAASGTGTYAT | | | |
| M-IV | *sul*1 | F-CGG CGT GGG CTA CCT GAA CG | 433 | 95°C for 15 min; 30 cycles of 95°C for 1 min, 66°C for 1 min and 72°C for 1 min; elongation step at 72°C for 10 min | [57] |
| | | R-GCC GAT CGC GTG AAG TTC CG | | | |
| | *sul*2 | F-CGG CGT GGG CTA CCT GAA CG | 721 | | [57] |
| | | R-GCC GAT CGC GTG AAG TTC CG | | | |
| | *sul*3 | F-CAACGGAAGTGG GCGTTG TGGA | 244 | | [57] |
| | | R-GCT GCA CCA ATT CGC TGAACG | | | |
| M-V | *tet*(A) | F-GGC GGTCTT CTT CAT CATGC | 502 | 94°C for 15 min; 30 cycles of 94°C for 1 min, 63°C for 1 min and 72°C for 1 min; elongation step at 72°C for 10 min | [57] |
| | | R-CGG CAG GCA GAG CAA GTAGA | | | |
| | *tet*(B) | F-CGC CCA GTG CTG TTG TTGTC | 173 | | [57] |
| | | R-CGC GTT GAG AAG CTG AGG TG | | | |
| | *tet*(C) | F-GCT GTAGGCATAGGCTTGGT | 888 | | [57] |
| | | R-GCC GGA AGC GAG AAGAATCA | | | |
| | *str*A/*str*B | F-ATGGTGGACCCTAAAACTCT | 893 | | [57] |
| | | R-CGTCTAGGATCGAGACAAAG | | | |

## Antimicrobial Susceptibility Testing (AST)

The Kirby-Bauer disc diffusion method was used to determine the AMR profile of all isolates, according to the Clinical and Laboratory Standards Institute's standards [50]. A panel of 16 antimicrobials representing 10 different classes were selected for AST consisting of aminoglycosides: amikacin (AK, 30μg), gentamicin (CN, 10μg), streptomycin (S, 10μg); carbapenem: meropenem (MEM, 10μg); cephalosporin/beta-lactam antibiotics: ceftriaxone (CRO, 30μg), cefotaxime (CT, 10μg), ceftazidime (CAZ, 30μg), aztreonam (ATM, 30μg); beta-lactamase inhibitors: amoxicillin–clavulanate (AMC, 30μg); penicillins: ampicillin (AMP, 10μg); macrolides: azithromycin (AZM, 15μg); quinolones/fluoroquinolones: ciprofloxacin (CIP, 5μg), nalidixic acid (NA, 30μg); folate pathway inhibitors: sulfamethoxazole-trimethoprim (SXT, 25μg); tetracycline: tetracycline (TE, 10μg); phenicols: chloramphenicol (C, 30μg). The isolates which were resistant to three or more classes of antibiotics were regarded as MDR [51]. The intermediate isolates were considered resistant as the acquisition and transition from susceptible to resistance had already begun [52]. The positive control was used as *Escherichia coli* ATCC 25922. The multiple antibiotic resistance (MAR) index was calculated and interpreted using a proven method [53, 54].

## MAR index calculation

Multiple Antibiotic Resistance (MAR) indexing has been considered as the cost effective and valid method for source tracking of a bacteria. MAR index is calculated as the ratio of number of resistant antibiotics to which organism is resistant to total number of antibiotics to which organism is exposed [55]. MAR index values larger than 0.2 indicate the organism is highly resistant where antibiotics are often used.

## PCR detection of AMR genes

The phenotypically resistant *Salmonella* isolates were screened by PCR for the detection of 14 antibiotic resistance genes, comprising of 7 β-lactamase genes (*bla*TEM, *bla*SHV, *bla*OXA, *bla*CTX-M-1, *bla*CTX-M-2, *bla*CTX-M-9 and *bla*CTX-Mg8/25*)*, 3 tetracycline resistant genes (*tet*A, *tet*B and *tet*C), 3 sulfonamide resistant genes (*sul*1, *sul*2 and *sul*3) and single streptomycin resistant gene (*str*A/B). For β-lactam gene, two cycles of multiplex PCR (M-II & M-III) were carried out following the proven method of Dallenne et al. [56]. Consecutively, two cycles of multiplex PCR (M-IV & M-V) were performed to detect the resistance genes for sulfonamide, tetracycline and streptomycin in consistent with the established method [57]. PCR reaction mixture, as well as gel electrophoresis was done in alignment with the procedures applied for the detection of *Salmonella enterica* serovars in this study. The primers used to detect resistance genes are presented in Table 1.

## PCR detection of virulence genes in *Salmonella* isolates

All *Salmonella* isolates were screened for the determination of eight important virulent genes encoding *inv*A, *agf*A, *Ipf*A, *hil*A, *siv*H, *sef*A, *sop*E and *spv*C. The PCR was executed in single reactions following previously used specific primers and thermal profiles [37, 59–64]. PCR reaction mixture, as well as gel electrophoresis was done in alignment with the procedures applied for the detection of *Salmonella enterica* serovars in this study. The reference positive control (*S*. Typhimurium ATCC 14028 and *S*. Enteritidis ATCC 13076) and negative control (*E. coli* ATCC 25922) were used for validation. The primers used in this study are presented in Table 2.

**Table 2. Primers used to detect virulence gene in *Salmonella* isolates.**

| PCR | Target Gene | Sequence | Amplicon size (bp) | Thermal Profile | References |
|---|---|---|---|---|---|
| P-1 | *agf*A | F-TCCACAATGGGGCGGCGGCG | 350 | 94˚C for 1 sec; 58˚C for 1 sec; 74˚C 21 sec | [59] |
| | | R-CCTGACGCACCATTACGCTG | | | |
| P-2 | *Ipf*A | F-CTTTCGCTGCTGAATCTGGT | 250 | 94˚C for 1 sec; 55˚C for 1 sec; 74˚C 21 sec | [37] |
| | | R-CAGTGTTAACAGAAACCAGT | | | |
| P-3 | *hil*A | F-CTGCCGCAGTGTTAAGGATA | 497 | 94˚C for 120 sec; 62˚C for 1 min; 72˚C 1 min | [60] |
| | | R-CTGTCGCCTTAATCGCATGT | | | |
| P-4 | *siv*H | F-GTATGCGAACAAGCGTAACAC | 763 | 94˚C for 30 sec; 56˚C for 45 sec; 72˚C 45 sec | [61] |
| | | R-CAGAATGCGAATCCTTCGCAC | | | |
| P-5 | *sef*A | F-GATACTGCTGAACGTAGAAGG | 488 | 94˚C for 1 sec; 56˚C for 1 sec; 74˚C 21 sec | [62] |
| | | R-GCGTAAATCAGCATCTGCAGTAGC | | | |
| P-6 | *sop*E | F-GGATGCCTTCTGATGTTGACTGG | 398 | 94˚C for 1 min; 55˚C for 1 min; 72˚C for 1 min | [63] |
| | | R-ACACACTTTCACCGAGGAAGCG | | | |
| P-7 | *spv*C | F-CCCAAACCCATACTTACTCTG | 669 | 93˚C for 1 min; 42˚C for 1 min; 72˚C for 2 min | [64] |
| | | R-CGGAAATACCATCTACAAATA | | | |

## Statistical analysis

The antimicrobial susceptibility data was presented in Excel sheets (MS-2016) and analyzed with SPSS software (SPSS-24.0). The prevalence was calculated using descriptive analysis and the Chi-square test was applied to determine the level of significance. Statistical significance was determined by a p-value less than 0.05 (p <0.05).

## Results

### Prevalence of NTS *enterica* serovars

Of all 870 samples, 165 (18.96%) were positive for *Salmonella*. The prevalence of *Salmonella* was found 20% (58 in 290) in CDW, 19.31% (56 in 290) in CBS, and 17.58% (51 in 290) in KS. Meanwhile, the MDR *Salmonella* was found to be 72.41% (42 in 58), 73.21% (41 in 56), and 68.62% (35 in 51) in CDW, CBS, and KS, respectively. The overall prevalence of *S*. Typhimurium, *S*. Enteritidis, and untyped *Salmonella* was found to be 8.96%, 1.6%, and 8.38%, respectively along with an overall MDR of 71.41%. The prevalence of NTS Typhimurium, Enteritidis and untyped *Salmonella* were found to be 7.93% (23 in 290), 1.72% (5 in 290), and 10.34% (30/290) in CDW, respectively. Likewise, the prevalence of NTS Typhimurium, Enteritidis, and untyped *Salmonella* was found to be 10.34% (30 in 290), 2.06% (6 in 290), and 6.89% (20/290) in CBS, respectively. Similarly, the prevalence of NTS Typhimurium, Enteritidis, and untyped *Salmonella* was represented 8.62% (25 in 290), 1.03% (3 in 290), and 7.93% (23 in 290) in KS, correspondingly.

### Phenotypic resistance patterns of NTS isolates

AST result in CDW revealed the highest resistance to ciprofloxacin (68.95%) followed by nalidixic acid (62.06%), tetracycline (60.33%), ampicillin, (58.61%) and streptomycin (56.88%); moderate resistance to gentamicin (39.64%), amoxicillin-clavulanate (31.92%), sulfamethoxazole-trimethoprim, (27.58%) and chloramphenicol (20.67%). On the contrary, low resistance was observed to azithromycin, amikacin and meropenem, respectively. Third-generation cephalosporins (ceftriaxone, cefotaxime, ceftazidime, and aztreonam) were found almost sensitive to all *Salmonella* isolates recovered from CDW (Table 3). Consequently, AST result of CBS showed higher resistance to streptomycin (64.28%) followed by ciprofloxacin (62.49%),

**Table 3. Phenotypic resistance pattern of NTS *enterica* serovars in carcass dressing water (CDW).**

| Serovar | Antimicrobial Resistance (%) | | | | | | | | | | | | | | | |
|---|---|---|---|---|---|---|---|---|---|---|---|---|---|---|---|---|
| | CIP | S | AMP | TE | NA | CN | SXT | AMC | C | AZM | AK | MEM | ATM | CRO | CT | CAZ |
| *S.* Typhimurium | 34.49 | 27.58 | 31.04 | 34.48 | 29.31 | 27.59 | 5.17 | 18.97 | 6.89 | 3.45 | 0 | 1.72 | 0 | 0 | 3.45 | 0 |
| *S.* Enteritidis | 6.89 | 3.44 | 5.17 | 3.44 | 5.17 | 3.44 | 5.17 | 3.44 | 1.72 | 0 | 1.72 | 0 | 0 | 0 | 0 | 0 |
| Untyped *Salmonella* | 27.58 | 25.89 | 22.41 | 22.42 | 27.58 | 8.62 | 17.24 | 8.62 | 12.07 | 3.44 | 1.72 | 1.72 | 0 | 0 | 3.44 | 0 |
| Overall Resistance | 68.96 | 56.9 | 58.62 | 60.34 | 62.06 | 39.65 | 27.58 | 31.03 | 20.68 | 6.89 | 3.44 | 3.44 | 0 | 0 | 6.89 | 0 |

ampicillin (62.27%), tetracycline (60.7%), nalidixic acid (53.56%), and gentamicin (53.56%); moderate resistance (14.27%-46.41%) was recorded for sulfamethoxazole-trimethoprim, amoxicillin-clavulanate, chloramphenicol, and azithromycin. Besides, complete sensitivity was found in all third-generation cephalosporins, including carbapenem (ceftriaxone, cefotaxime, ceftazidime, aztreonam, and meropenem) (Table 4). Successively, AST result of KS exhibited higher resistance to ciprofloxacin (64.69%), ampicillin (64.69%), streptomycin (64.7%), nalidixic acid (58.81%), and tetracycline (54.89%); moderate resistance was recorded to gentamicin (47.05%), sulfamethoxazole-trimethoprim (47.05%), amoxicillin-clavulanate (27.44%) and chloramphenicol (21.56%). On the contrary, very low resistance or almost sensitivity were observed to azithromycin, amikacin, meropenem, and third-generation cephalosporins (ceftriaxone, cefotaxime, ceftazidime, and aztreonam) (Table 5). There was harmony and synergy among the phenotypic resistance patterns of CDW, CBS, and KS. The AST pattern of ciprofloxacin in CDW was significantly higher compared to CBS ($p < 0.05$). Similarly, the AST pattern of gentamicin in CBS was significantly higher compared to CDW and KS ($p < 0.05$). A statistical association of phenotypic resistance patterns was found among the carcass treatment water, cutting board swabs, and knife swabs ($p<0.05$). The details of phenotypic and genotypic antimicrobial resistance data of *Salmonella* sevovars are presented in S1 Text.

## MAR index patterns of NTS isolates

The large phenotypic resistance pattern in CDW was found CIP-S-AMP-TE-NA-CN-AMC-SXT-CT-MEM while most one was CIP-S-AMP-TE-NA-CN-AMC. Similarly, the large phenotypic resistance pattern in CBS was found CIP-S-AMP-TE-NA-CN-AMC-AZM-SXT while the most one was CIP-S- CIP-S-AMP-TE-NA-CN-AMC. Likewise, the large phenotypic resistance pattern in KS was found CIP-S-AMP-TE-NA-AMC-SXT-CT--CAZ-CRO-ATM while the most common one was CIP-S-AMP-TE-NA-CN-AMC-C. The overall MAR index of more than 0.2 was found in 50%, 50%, 64.7% isolates of CDW, CBS and KS respectively. Besides, the highest MAR index value of 0.68, 0.62, and 0.56 was recorded in KS, CDW, and CBS, respectively. The complete sensitive isolates were identified at 1.39% (4 in 290), 2.06% (6 in 290), and 1.03% (3 in 290) in the CDW, CBS, and KS, respectively. The AMR patterns and MAR index of *Salmonella enterica* serovars are shown in S2 Text.

**Table 4. Phenotypic resistance pattern of NTS *enterica* serovars in chopping board swab (CBS).**

| Serovar | Antimicrobial Resistance (%) | | | | | | | | | | | | | | | |
|---|---|---|---|---|---|---|---|---|---|---|---|---|---|---|---|---|
| | CIP | S | AMP | TE | NA | CN | SXT | AMC | C | AZM | AK | MEM | ATM | CRO | CT | CAZ |
| *S.* Typhimurium | 39.28 | 41.07 | 42.85 | 42.86 | 41.07 | 39.29 | 23.21 | 23.22 | 10.72 | 10.72 | 0 | 0 | 0 | 0 | 0 | 0 |
| *S.* Enteritidis | 7.15 | 7.14 | 5.35 | 5.35 | 7.14 | 1.78 | 5.36 | 1.78 | 0 | 1.78 | 0 | 0 | 0 | 0 | 0 | 0 |
| Untyped *Salmonella* | 16.07 | 16.07 | 19.65 | 12.5 | 5.36 | 12.5 | 17.85 | 10.71 | 8.92 | 1.78 | 1.78 | 0 | 0 | 0 | 0 | 0 |
| Overall Resistance | 62.5 | 64.28 | 67.85 | 60.71 | 53.57 | 46.42 | 46.42 | 35.71 | 19.64 | 14.28 | 1.78 | 0 | 0 | 0 | 0 | 0 |

**Table 5. Phenotypic resistance pattern of NTS *enterica* serovars in knife swab (KS).**

| Serovar | Antimicrobial Resistance (%) | | | | | | | | | | | | | | | |
|---|---|---|---|---|---|---|---|---|---|---|---|---|---|---|---|---|
| | CIP | S | AMP | TE | NA | CN | SXT | AMC | C | AZM | AK | MEM | ATM | CRO | CT | CAZ |
| *S.* Typhimurium | 43.14 | 35.29 | 41.17 | 41.17 | 41.18 | 37.25 | 31.37 | 21.57 | 15.68 | 5.88 | 0 | 1.96 | 0 | 0 | 1.96 | 0 |
| *S.* Enteritidis | 0 | 1.96 | 3.93 | 3.92 | 5.88 | 0 | 1.96 | 1.96 | 1.96 | 0 | 1.96 | 1.96 | 0 | 0 | 0 | 0 |
| Untyped *Salmonella* | 21.56 | 27.45 | 19.6 | 9.8 | 11.76 | 9.8 | 13.72 | 3.92 | 3.92 | 1.96 | 1.96 | 1.96 | 5.88 | 3.92 | 5.88 | 5.88 |
| Overall Resistance | 64.7 | 64.7 | 64.7 | 54.9 | 58.82 | 47.05 | 47.05 | 27.45 | 21.56 | 7.84 | 3.92 | 5.88 | 5.88 | 3.92 | 7.84 | 5.88 |

## Genotypic resistance patterns of NTS isolates

All phenotypically resistant *Salmonella* isolates were screened by PCR for the detection of 14 antibiotic resistant genes encompassing 7 β-lactamase genes (*bla*TEM, *bla*SHV, *bla*OXA, *bla*CTX-M-1, *bla*CTX-M-2, *bla*CTX-M-9 and *bla*CTX-Mg8/25*)*, 3 tetracycline resistant genes (*tet*A, *tet*B and *tet*C), 3 sulfonamide resistant genes (*sul*1, *sul*2, and *sul*3) and single streptomycin resistant gene (*str*A/B) recovered from CDW (Table 6), CBS (Table 7) and KS (Table 8).

Out of seven, only one ESBL gene, *bla*TEM was detected with a prevalence rate of 62.06%, 69.62%, and 62.73% in CDW, CBS, and KS, respectively. Consecutively, out of three tetracycline resistant genes, only one *tet*A was identified with a prevalence level of 60.32%, 58.92%, and 58.81% in CDW, CBS, and KS, respectively. Sequentially, the prevalence of the *sul*1 gene was found 60.33%, 69.62%, and 49.01% in CDW, CBS, and KS, respectively. Furthermore, *sul*2 and *sul*3 were found in CBS with lower prevalence rate of 3.56% and 3.56%, respectively. Similarly, the *sul*3 gene was detected in KS with a prevalence rate of 17.62%. Moreover, the streptomycin resistance gene, *str*A/B was detected with a prevalence rate of 36.2%, 24.99%, and 31.36% in CDW, CBS, and KS, respectively. The detailed genotypic susceptibility pattern, including *Salmonella enterica* serovars and untyped *Salmonella* from three different sources, is presented in tabular form (Tables 6–8). The *sul*1gene in CBS was significantly higher compared to CDW and KS ($p < 0.05$). Similarly, the *str*A/B gene in CDW was significantly higher compared to KS and CBS ($p < 0.05$). A statistical association of genotypic resistance patterns was found among the carcass treatment water, cutting board swabs, and knife swabs ($p<0.05$).

## PCR detection of virulence genes for NTS isolates

All *Salmonella* isolates were screened by PCR to monitor eight common virulence genes namely *inv*A, *agf*A, *Ipf*A, *hil*A, *siv*H, *sef*A, *sop*E, and *spv*C. *S.* Enteritidis and untyped *Salmonella* isolates were found positive for all eight common virulence genes whereas *S.* Typhimurium harbored six virulence genes (Table 9). The analysis showed significantly higher associations among the virulence genes in CDW, CBS, and KS respectively ($p < 0.05$). The detail statistical analysis is given in S3 Text.

**Table 6. Genotypic resistance pattern of NTS *enterica* serovars in carcass dressing water.**

| Serovar | Genotypic resistance (%) | | | |
|---|---|---|---|---|
| | *bla*TEM | *Tet*A | *Sul*1 | *Str*A/B |
| *S.* Typhimurium | 39.65 | 36.2 | 34.48 | 13.79 |
| *S.* Enteritidis | 8.62 | 3.44 | 3.44 | 3.44 |
| Untyped *Salmonella* | 13.79 | 20.68 | 22.41 | 18.96 |
| Overall resistance | 62.06 | 60.32 | 60.33 | 36.20 |

**Table 7. Genotypic resistance pattern of NTS *enterica* serovars in chopping board swab.**

| Serovar | Genotypic resistance (%) | | | | | |
|---|---|---|---|---|---|---|
| | *bla*TEM | *Tet*A | *Sul*1 | *Sul*2 | *Sul*3 | *Str*A/B |
| *S.* Typhimurium | 42.85 | 41.07 | 42.85 | 1.78 | 1.78 | 14.28 |
| *S.* Enteritidis | 5.35 | 5.35 | 5.35 | 0 | 0 | 3.57 |
| Untyped *Salmonella* | 21.42 | 12.5 | 21.42 | 1.78 | 1.78 | 7.14 |
| Overall resistance | 69.62 | 58.92 | 69.62 | 3.56 | 3.56 | 24.99 |

**Table 8. Genotypic resistance pattern of NTS *enterica* serovars in knife swab.**

| Serovar | Genotypic resistance (%) | | | | |
|---|---|---|---|---|---|
| | *bla*TEM | *Tet*A | *Sul*1 | *Sul*3 | *Str*A/B |
| *S.* Typhimurium | 43.13 | 41.17 | 37.25 | 9.8 | 15.68 |
| *S.* Enteritidis | 3.92 | 3.92 | 1.96 | 0 | 0 |
| Untyped *Salmonella* | 15.68 | 13.72 | 9.8 | 7.84 | 15.68 |
| Overall resistance | 62.73 | 58.81 | 49.01 | 17.62 | 31.36 |

# Discussion

NTS *enterica* serovars isolated from chicken processing environments at wet markets in Bangladesh have only been reported in a few investigations. In our research, we discovered an overall prevalence of *Salmonella* 18.96% in poultry processing environmental samples such as CDW, CBS, and KS. Previously, in Bangladesh the prevalence of *Salmonella* was found to be present 23.33% in poultry slaughter specimens [23]; 26.6% in chicken cloacal swab, intestinal fluid, egg surface, hand wash, and soil of chicken market samples [65]; 25.35% in the chicken cloacal swab, eggshells, intestinal contents, liver swabs, broiler meat, and swabs of slaughterhouse [66]; 35% in broiler farms settings [31]; 23.53% in poultry samples [67]; 37.9% in poultry production settings [68]; 31.25% in broiler farm settings [69]; 42% in broiler chicken [70] and 65% in frozen chicken meat [71]. The prevalence of *Salmonella* was found to be 8.62% in broiler, 6.89% in sonali and 3.1% in native chicken cecal contents according to Siddiky et al. [32]. The prevalence of *Salmonella* isolates in our findings was consistent with earlier findings of Bangladesh.

**Table 9. Virulence genes distribution among the *Salmonella enterica* serovars.**

| Samples | Serovar | Prevalence of virulence genes (%) | | | | | | | |
|---|---|---|---|---|---|---|---|---|---|
| | | *Inv*A | *Agf*A | *Ipf*A | *Hil*A | *Siv*H | *Sop*E | *Sef*A | *Spv*C |
| CDW | *S.* Typhimurium | 100 | 100 | 100 | 100 | 100 | 100 | 0 | 0 |
| | *S.* Enteritidis | 100 | 100 | 100 | 100 | 100 | 100 | 100 | 100 |
| | Untyped *Salmonella* | 100 | 100 | 100 | 100 | 100 | 100 | 100 | 100 |
| CBS | *S.* Typhimurium | 100 | 100 | 100 | 100 | 100 | 100 | 0 | 0 |
| | *S.* Enteritidis | 100 | 100 | 100 | 100 | 100 | 100 | 100 | 100 |
| | Untyped *Salmonella* | 100 | 100 | 100 | 100 | 100 | 100 | 100 | 100 |
| KS | *S.* Typhimurium | 100 | 100 | 100 | 100 | 100 | 100 | 0 | 0 |
| | *S.* Enteritidis | 100 | 100 | 100 | 100 | 100 | 100 | 100 | 100 |
| | Untyped *Salmonella* | 100 | 100 | 100 | 100 | 100 | 100 | 100 | 100 |

In Ethiopian butcher shops, the overall prevalence of *Salmonella* was determined to be 17.3%. *Salmonella* was found in KS, CBS, hand washings, and meat, which is consistent with our findings [72]. The study based on the wet market conducted in India revealed the prevalence of *Salmonella* of 14.83% in the chicken meat shops [73]; 19.04% in retail chicken stores [74]; and 23.7% in white and red meat in local markets [75]. A study demonstrated the high prevalence of *Salmonella* (88.46%) in poultry processing and environmental samples obtained from wet markets and small-scale processing plants in Malaysia [39]. *Salmonella* was found 35.5% and 50% in broiler carcasses at wet markets and processing plants, respectively, according to Rusul et al. [76]. Furthermore, in Penang, Malaysia, the overall incidence of *Salmonella* serovars was found to be 23.5% in ducks, duck raising, and duck processing environments [53]. Furthermore, our findings were connected to the recent frequency of *Salmonella* both at home and abroad. In our investigation, the prevalence of NTS Typhimurium was determined to be 7.93%, 10.34%, and 8.62% in CDW, CBS, and KS, respectively. Similarly, it was noted that the occurrence of *S.* Enteritidis was 1.72%, 2.06%, and 1.03% in CDW, CBS, and KS respectively. Siddiky et al. [32] found the overall prevalence of *S.* Typhimurium and *S.* Enteritidis at the rate of 3.67% and 0.57% in chicken cecal contents. There was a link between the prevalence of *Salmonella enterica* serovars in caecal content and environmental samples. Thung et al. [77] found *S.* Enteritidis and *S.* Typhimurium in raw chicken meat at retail markets in Malaysia, with prevalence rates of 6.7% and 2.5%, respectively. The major *Salmonella* enterica serovars in our investigation was *S.* Typhimurium, which was similar with the findings of McCrea et al. [78], who identified *S.* Typhimurium as the major *Salmonella* serovars from a California poultry market. According to Saitanu et al. [79], *S.* Typhimurium (5.5%) was the most common serotype in duck eggs in Thailand. The studies conducted in Bangladesh, *S.* Typhimurium was found to be 15.91% in broiler production systems [80]; 85% in broiler farm samples [31] and 5% in commercial layer farm settings [81]. *S.* Typhimurium was found to be more common in our study, which corresponds to findings both at home and overseas. Consecutively, *S.* Typhimurium and *S.* Enteritidis were isolated from raw chicken meat at retail markets in Malaysia [77]; higher prevalence of *S.* Enteritidis (21.9%) and *S.* Typhimurium (9.4%) were isolated from chicken in Turkey [82]; *Salmonella enterica* serovars were identified in backyard poultry flocks in India [83]; *S.* Enteritidis and *S.* Typhimurium recovered from chicken meat in Egypt [84]. Suresh et al. [22] recovered *S.* Typhimurium and *S.* Enteritidis in large proportions from various poultry products in India, compared to other serovars. Furthermore, China and some European countries detected *S.* Enteritidis and *S.* Typhimurium from catering points and meat of pork, chicken and duck as the most prevalent serotypes [85, 86].

In our study, NTS *enterica* serovars Typhimurium and Enteritidis along with untyped *Salmonella* was found higher resistance to ciprofloxacin, streptomycin, gentamicin, ampicillin, tetracycline, and nalidixic acid; moderate resistance to sulfamethoxazole-trimethoprim, amoxicillin-clavulanate, chloramphenicol, and azithromycin. Alam et al. [31] found a high percentage of *Salmonella* resistance to tetracycline, ampicillin, streptomycin, and chloramphenicol (77.1% to 97.1%). Furthermore, according to Parvin et al. [71], *Salmonella* has the highest resistance to oxytetracycline (100%), followed by trimethoprim-sulfamethoxazole (89.2%), tetracycline (86.5%), nalidixic acid (83.8%), amoxicillin (74.3%), and pefloxacin (74.3%). Mridha et al. [69] shown higher to moderate resistance of the isolates of *Salmonella* to erythromycin, tetracycline, amoxicillin, and azithromycin. Sequentially, Sobur et al. [87] found higher resistance of *Salmonella* to tetracycline, ciprofloxacin, and ampicillin. *Salmonella* isolated from the feces of chickens, ducks, geese, and pigs has been reported to be resistant to nalidixic acid (48.8%), tetracycline (46.9%), ampicillin (43.2%), streptomycin (38.3%), and trimethoprim/sulfamethoxazole (33.3%), respectively [88–90]. It was found that *Salmonella* was highly

resistant to ciprofloxacin (77%), sulfisoxazole (73%) and ampicillin (55.6%) in chicken hatcheries in China [91]; highly resistance to tetracycline and ampicillin in wet markets, Thailand [41]; higher resistance to nalidixic acid (99.5%), ampicillin (87.8%), tetracycline (51.9%), ciprofloxacin (48.7%), and trimethoprim-sulfamethoxazole (48.1%) in broiler chickens along the slaughtering process in China [92]. *S. enterica* serovar Typhimurium was found to be resistance to ampicillin, tetracycline, and sulphamethoxazole isolated from chicken farms in Egypt [93]. The NTS *enterica* serovars was found to be higher resistance to ampicillin (95.71%), ciprofloxacin (82.86%), tetracycline (100%), and nalidixic acid (98.57%) in retail chicken meat stores in northern India [73]. Siddiky et al. [32] found that *S.* Typhimurium had the highest resistance to ciprofloxacin (100%) and streptomycin (100%) followed by tetracycline (86.66%), nalidixic acid (86.66%), gentamicin (86.66%), ampicillin (66.66%), and amoxicillin–clavulanate (40%) in in broiler chickens. Furthermore, Siddiky et al. [32] reported the highest resistance pattern of *S.* Typhimurium to ciprofloxacin (100%) and streptomycin (100%) followed by tetracycline (86.66%), nalidixic acid (86.66%), gentamicin (86.66%), ampicillin (66.66%) and amoxicillin–clavulanate (40%) in broiler chicken. Similarly, Siddiky et al. [32] identified the maximum resistance of the *S.* Enteritidis to streptomycin (100%) followed by ciprofloxacin (80%), tetracycline (80%), gentamicin (80%), and moderate resistance to amikacin (20%), amoxicillin–clavulanate (20%), azithromycin ((20%), and sulphamethazaxole-trimethoprim (20%) in broiler chicken. There was a substantial correlation and congruence with phenotypic resistance patterns of *Salmonella enterica* serovars both at home and abroad.

According to Mishra et al. [94], MAR index of 0.2 or higher indicates high risk sources of contamination, MAR index of 0.4 or higher is associated with fecal source of contamination. Thenmozhi et al. [95], also states that MAR index values > 0.2 indicate existence of isolate from high risk contaminated source with frequency use of antibiotics while values ≤ 0.2 show bacteria from source with less antibiotics usage. High MAR indices mandate vigilant surveillance and remedial measures. In this study, the overall MAR index of more than 0.2 was found in 50%, 50%, and 64.7% isolates of CDW, CBS, and KS respectively. Besides, the highest MAR index value of 0.68, 0.62, and 0.56 was recorded in KS, CDW, and CBS respectively.

A single beta-lactam-resistant *bla*TEM gene was discovered in all three categories of samples with a different frequency rate in our analysis, out of seven. In CDW, CBS, and KS, the prevalence of *bla*TEM was found to be 62.06%, 69.62%, and 62.73%, respectively. Ahmed et al. [96] detected a higher prevalence of *bla*TEM mediated ESBL gene among *Salmonella* isolated from humans in Bangladesh. Yang et al. [97] identified *bla*TEM, a gene encoded for beta-lactamases resistance, in 51.6% resistant *Salmonella* isolates. According to Aslam et al. [98], the *bla*TEM gene was found in 17% of *Salmonella* isolates from retail meats in Canada. Lu et al. [99] detected only 81.2% *bla*TEM gene, while *bla*CTX-M could not be detected in any of the examined isolates. Similarly, Van et al. [100] found only the *bla*TEM gene in *E. coli* recovered from raw meat and shellfish in Vietnam. The emergence of *bla*TEM mediated ESBL producing NTS *enterica* serovars indicated the use of beta-lactam antibiotics in poultry farming practices. Siddiky et al. [32] detected only the *bla*TEM gene from chicken cecal contents and Xiang et al. [101] reported plasmid-borne and easily transferable *bla*OXA-1 and *bla*TEM-1 genes. Consecutively, Suresh et al. [102] detected *bla*TEM as the predominant gene in food of animal origin in India. The *bla*TEM gene's results were consistent with and related to earlier findings. In our analysis, just one tetracycline resistance gene, *tet*A, was found in carcass dressing water, chopping board swab, and knife swab, with prevalence rates of 60.32%, 58.92%, and 58.81%, respectively. The *sul*1, *sul*2, and *sul*3 genes were also found in NTS *enterica* serovars, with *sul*1 being the most common. Similarly, the streptomycin resistance gene (*str*A/B) was found in NTS serovars with a high prevalence rate. Arkali and Çetinkaya [82] detected 58% positive *sul*1 gene from the *Salmonella* isolates of chickens in eastern Turkey. Consequently, Jahantigh et al.

[103] detected the most prevalent *tet*A gene from broiler chickens in Iran. Successively, Vuthy et al. [90] discovered *bla*TEM, *tet*A and *str*A/B genes from the chicken food chain, while Sin et al. [104] isolated *tet*A and *sul*1 genes from chicken meat in Korea. Zhu et al. [92] isolated beta lactam (*bla*TEM), tetracycline resistant (*tet*A, *tet*B, *tet*C) and sulfonamide resistant (*sul*1, *sul*2 and *sul*3) genes with prevalence of 94.6%, 85.7%, and 97.8% in *Salmonella* isolated from the slaughtering process in China. El-Sharkawy et al. [93] who revealed *bla*TEM, *tet*A, *tet*C, *sul*1, and *sul*3 genes from *S*. Enteritidis isolates at a chicken farm in Egypt. Doosti et al. [105] detected *str*A/B (37.6%) from *S*. Typhimurium isolates at poultry carcasses in Iran. Sharma et al. [73] detected the most predominant *tet*A and *bla*TEM genes in NTS isolated from retail chicken shops in India. Continually, Alam et al. [31] revealed *tet*A (97.14%) and *bla*TEM-1 (82.85%) genes in broiler farms, whilst Siddiky et al. [32] detected *tet*A, *sul*1, and *str*A/B genes in chicken cecal content in Bangladesh. The genotypic resistance patterns were well matched with previous findings at home and abroad.

In our study, harmonic and proportioned correlations were existent between genotypic and phenotypic resistance decoration. These findings were in agreement and alignment with the observations of previous studies conducted across the globe [31, 32, 92]. However, sometimes the phenotypic and genotypic resistance pattern were not found to be similar. This may be due to source and concentration of the antibiotic disk, source of primers, concentration of inoculum, facilities of the laboratory and the capacity and skills of the laboratory personnel [32]. Previous research findings supported the disagreement between genotypic and phenotypic resistance patterns [100, 106].

In our study, MDR *Salmonella* embedded mostly with ciprofloxacin, streptomycin, tetracycline, ampicillin, gentamicin, and nalidixic acid, probably due to the common and frequent use of these antibiotics in poultry production settings in Bangladesh [29, 107]. CLSI [50] reported *Salmonella* has become naturally resistant to first and second-generation cephalosporins and aminoglycosides. The MDR along with higher resistance to ciprofloxacin is very alarming in human treatment as WHO recommended ciprofloxacin, a first-line drug treatment of intestinal infections. Besides, watch group ciprofloxacin had higher resistance and azithromycin had moderate resistance, reflects the severity of the resistance pattern of *Salmonella* serovars [108]. Furthermore, higher resistance to tetracycline indicated the massive use of therapeutic and growth enhancers in poultry production [109]. The emergence of *bla*TEM mediated ESBL producing *Salmonella enterica* serovars indicated the use of beta-lactam antibiotics in the poultry production cycle. Moreover, ESBL is usually encoded by large plasmids that are transferable from strain to strain and between bacterial species [96, 110].

Virulence gene analysis indicated *S*. Enteritidis and untyped *Salmonella* isolates carried eight virulence genes, including two types of *Salmonella* pathogenicity islands (SPI-1 and SPI-2) and many adhesion-related virulence genes. Virulence genes along with the MDR resistance pattern would accelerate the infectivity of *Salmonella* isolates [35]. The emergence of antibiotic resistance of *Salmonella* isolates depends on their genetic and pathogenicity mechanisms, which can enhance their survivability by preserving drug resistance genes [98]. The virulence gene was found to be more prevalent in *S*. Enteritidis and untyped *Salmonell*a isolates compared to *S*. Typhimurium isolates. Six common virulence genes (*inv*A, *agf*A, *Ipf*A, *hil*A, *siv*H, and spvC) were detected in all isolates of *Salmonella*, which was incompatible with prior findings around the world [111–114]. Moreover, *sef*A and *spv*C genes were detected in *S*. Enteritidis and untyped *Salmonella* isolates, whilst none of the *S*. Typhimurium isolates carried *sef*A and *spv*C genes. Alike findings have been recorded previously by researchers [112]. The higher occurrence of *sef*A in *S*. Enteritidis was compatible with prior findings [111, 114], and *sef*A was somewhat considered a target gene for encountering *S*. Enteritidis through the PCR method [111]. Successively, the *inv*A gene was the most common and virulent gene present in all

*Salmonella* isolates and was considered as a target gene for identifying *Salmonella* species [33, 115]. Continually, the *hil*A gene played a key role in exaggerating *Salmonella* virulence by stimulating the expression of invasion into the cell [116, 117]. Furthermore, the virulence genes *inv*A and *hil*A could be considered target genes for rapid and reliable detection of *Salmonella* through the PCR method. The higher occurrence of *lpf*A, *agf*A, and *sop*E were consistent with previous research results [38, 118]. The occurrence of the *sop*E gene (100%) in *S*. Enteritidis was correlated with earlier studies [119]. Further, the *agf*A gene is liable for biofilm development along with adhesion to cells during the infection process [120]. In our study, the plasmid-mediated *spv*C virulence gene was detected in *S*. Enteritidis and untyped *Salmonella* isolates, which have similarities to earlier observations [111, 121, 122]. It was previously found *S*. Enteritidis had 92% *spv*C gene while *S*. Typhimurium had only 28% and *S*. Hadar had none [112]. The higher prevalence of major virulence genes indicated the pathobiology as well as the public health implications of the serovars. Moreover, all *Salmonella enterica* isolates were found to be highly invasive and enterotoxigenic, which had a significant public health impact. This was the first-ever attempt to determine a wider range of *Salmonella* virulence genes from poultry processing environments at wet markets in Dhaka, Bangladesh.

Our results demonstrated that wet markets where chicken has been slaughtered and processed could spread and harbour NTS *enterica* serovars. Many studies have shown that cross-contamination of poultry could occur during processing and skinning in wet markets due to poor sanitary and hygienic measures [39]. In wet markets, the sources of contamination may be vendors, chopping board swab, knife swab, carcass dressing water, defeathering machines, scalding water, tanks, floors, drains, and work benches, etc. [39]. Free roaming MDR NTS serovars in poultry processing environments could facilitate release into the food chain, agricultural goods, and human populations in wet markets. Furthermore, *Salmonella* serovars were able to survive longer in soil-formed biofilms, and these biofilms were protected from detergents and sanitizers [123]. Therefore, cleaning and sterilization of the knife, chopping board, and frequent change of carcass dressing water are crucial to reduce the burden of horizontal transmission of *Salmonella* enterica in wet markets. The root causes indicated that MDR and highly pathogenic NTS *enterica* have emerged in poultry due to the irrational use of antimicrobials in farming practices.

## Conclusion

The higher prevalence of multiple virulence and multidrug resistant NTS *enterica* serovars in the poultry processing environments drew public health attention. Chicken carcasses are dressed and processed in the open environment of the wet market, which exacerbated the spread of pathogens. The unclean and utilized utensils such as chopping boards and knives were used for chicken processing and cutting. Furthermore, unclean and dirty water were used frequently for chicken carcass dressing and washing. Numerous risk factors are prevailing at wet markets that can trigger the spread and contamination of NTS serovars. The wet market can be considered a hotspot for harboring NTS serovars which can easily anchor in the food chain as well as human health. The hidden source of these MDR pathogens was undoubtedly chickens, which indicates that there was more therapeutic and preventive exposure to antibiotics during the production cycle. The clean, hygienic and ambient poultry processing environments with good carcass processing practices might reduce the spread of contamination at wet markets. Besides, prudent and judicious use of antimicrobials has to be ensured in farming practices, as poultry farming is considered a fertile ground for the use of antimicrobials. This study could address the potential risk associated with the spread of NTS with multidrug

resistance to humans as well as highlight the need to implement a strict hygiene and sanitation standards in local wet markets.

## Supporting information

**S1 Text. Phenotypic and genotypic antimicrobial resistance data.**
(DOCX)

**S2 Text. AMR patterns and MAR index of *Salmonella enterica* serovars.**
(DOCX)

**S3 Text. Statistical analysis data.**
(DOCX)

## Acknowledgments

The author is very grateful for the logistics and technical support by the project "Combating the threat of antibiotic resistance and zoonotic diseases to achieve Bangladesh's GHSA" during sample collection. We would like to thank Dr. Mohammod Kamruj Jaman Bhuiyan, Department of Agricultural and Applied Statistics, Bangladesh Agricultural University, Mymensingh-2202 for statistical analysis.

## Author Contributions

**Conceptualization:** Nure Alam Siddiky, Shahidur Rahman Khan, Mohammed A. Samad.

**Data curation:** Nure Alam Siddiky, Samun Sarker.

**Formal analysis:** Nure Alam Siddiky.

**Methodology:** Nure Alam Siddiky, Samun Sarker.

**Software:** Nure Alam Siddiky, Samun Sarker.

**Supervision:** Shahidur Rahman Khan, Tanvir Rahman, Abdul Kafi, Mohammed A. Samad.

**Validation:** Tanvir Rahman, Abdul Kafi.

**Visualization:** Nure Alam Siddiky, Samun Sarker.

**Writing – original draft:** Nure Alam Siddiky, Samun Sarker.

**Writing – review & editing:** Shahidur Rahman Khan, Tanvir Rahman, Abdul Kafi, Mohammed A. Samad.

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

33451238

