## [Decision Letter · Decision Letter 0]

18 Aug 2021

PONE-D-21-19124

Virulence and antimicrobial resistance profile of non-typhoidal Salmonella enterica serovars recovered from poultry processing environments at wet markets in Dhaka, Bangladesh

PLOS ONE

Dear Dr. Samad,

Thank you for submitting your manuscript to PLOS ONE. After careful consideration, we feel that it has merit but does not fully meet PLOS ONE’s publication criteria as it currently stands. Therefore, we invite you to submit a revised version of the manuscript that addresses the points raised during the review process.

We look forward to receiving your revised manuscript.

Kind regards,

Kumar Venkitanarayanan, DVM, Ph.D.

Academic Editor

PLOS ONE

Journal Requirements: 

3. We note that Figure (1) in your submission contain [map/satellite] images which may be copyrighted. All PLOS content is published under the Creative Commons Attribution License (CC BY 4.0), which means that the manuscript, images, and Supporting Information files will be freely available online, and any third party is permitted to access, download, copy, distribute, and use these materials in any way, even commercially, with proper attribution. For these reasons, we cannot publish previously copyrighted maps or satellite images created using proprietary data, such as Google software (Google Maps, Street View, and Earth). For more information, see our copyright guidelines: http://journals.plos.org/plosone/s/licenses-and-copyright.

1. You may seek permission from the original copyright holder of Figure (1) to publish the content specifically under the CC BY 4.0 license.  

Reviewers' comments:

Reviewer's Responses to Questions

**Comments to the Author**

1. Is the manuscript technically sound, and do the data support the conclusions?

Reviewer #1: Yes

Reviewer #2: Partly

2. Has the statistical analysis been performed appropriately and rigorously? 

Reviewer #1: No

Reviewer #2: Yes

3. Have the authors made all data underlying the findings in their manuscript fully available?

Reviewer #1: Yes

Reviewer #2: Yes

4. Is the manuscript presented in an intelligible fashion and written in standard English?

Reviewer #1: No

Reviewer #2: No

5. Review Comments to the Author

Reviewer #1: This manuscript outlines the prevalence, serovar identification, virulence and antimicrobial resistance profile typing of non-typhoidal Salmonella enterica isolated from poultry processing environments of wet markets in Dhaka. Specifically, the samples were collected from dressing water, chopping board and knives, which are potential abiotic routes for Salmonella entry during preharvest processing of poultry meat. The information provided by the authors would be beneficial to categorize the prevalence of Salmonella in poultry production systems and consider to use to data to enforce food safety precautionary measures to reduce the incidence of foodborne Salmonella outbreaks. This information is highly relevant for food safety and public health experts.

However, the manuscript requires a lot editing and revision as there were a lot of issues related grammatical structure and phrasing in the scientific context. I would highly recommend that the authors make of use of professional language editing and copyediting services to improve the presentation of the manuscript overall.

In addition, the authors should also improve the description in the statistical analysis section. In this section, the authors may consider to describe this in the following order - the kind of data generated from this study (percent prevalence, MAR index, etc., ), the tests used, the way the data is expressed (mean +/- SD or SE), statistical test chosen for analyzing each data type, the level of significance and then finally the statistical software that was used.

In the methods section, I would suggest that the authors describe more about how the MAR index calculation is performed apart from providing the reference for the method. Also, how was this data analyzed in the current study?

Reviewer #2: General Comments: The authors did a great job in determining the virulence and pathogenicity of Salmonella isolated from wet markets. however, the manuscript was poorly written with many sentences either copied and pasted or rearranged from another article published by the same author.

Specific comments:

1. Introduction:

a. There were numerous grammatical errors throughout the manuscript

b. Several statements were mentioned repeatedly. For example: Poultry as a source of Salmonella infections has been repeatedly mentioned

c. Lines 92 -103: Plagiarism noted from another article from the same author

2. Materials and Methods

a. Please explain the number of samples collected per site. Were samples collected on a single visit for each of the site?

b. Figure 1, Title should explain the inset and the pointers marked in the figure

c. Plagiarism- Lines 149-150, 156 – 160

d. Table 1: Please include the target gene names

e. Please explain MAR

3. Results

a. It would be better to represent Figure 2A-C as a table with the percentage of resistance.

b. sul3 gene was detected in both KS and CBS. Please correct line 279

c. Please explain “There was a strong correlation exist among the virulence genes in CDW, CBS, and KS (p < 0.05)” (line 293). It would be better to give figure 4 as a table as it is difficult to understand the results and, to identify the correlation between samples.

4. Discussion

a. Please rewrite the discussion section to contain only relevant data pertaining to the results.

b. 416- 419-Plagiarism noted

c. Please add reference for the statement Line 475

d. Line 344 – “Furthermore, China and some European countries detected S. Enteritidis and S. Typhimurium as the most prevalent serotypes” Did not explain from where they detected the serotypes.

6. PLOS authors have the option to publish the peer review history of their article (what does this mean?). If published, this will include your full peer review and any attached files.

Reviewer #1: No

Reviewer #2: No

---

## [Author Response · Author response to Decision Letter 0]

7 Oct 2021

Editor Comments 

Please ensure that your manuscript meets PLOS ONE's style requirements, including those for file naming

Response: The style of PLOS ONE has been followed as per given guidelines of the journal.

PLOS requires an ORCID ID for the corresponding author

Response: The ORCID ID for corresponding author is included (0000-0001-7329-7325)

Copyright for Figure 1(Map)

Response: The map (figure 1) is created for this study by using longitude and latitude data. Indeed, the copyright is ours.

Reviewer #1: This manuscript is suitable for publication following suggested modifications 

Manuscript requires a lot of editing and revision as there were a lot of issues related grammatical structure and phrasing in the scientific context.

Response: The professional linguistic editing has been made in the whole manuscript where it is appropriate. Necessary correction made and highlighted in the whole manuscript. 

Improve the description in the statistical analysis section

Response: The description of the statistical section has been improved as suggested by the reviewer (Line number: 226-229).

MAR index calculation in this study

Response: The detail method of MAR index calculation is included in the material & method section (Line number: 194-199).

Reviewer #2: This manuscript is suitable for publication following suggested modifications

The manuscript was poorly written with many sentences either copied and pasted or rearranged from another article published by the same author

Response: The English editing has been done in the whole manuscript and rewrite/ rephrase the sentences which is highlighted by the yellow color in the text.

There were numerous grammatical errors throughout the manuscript

Response: The grammatical error has corrected throughout the manuscript.

Several statements were mentioned repeatedly. For example: Poultry as a source of Salmonella infections has been repeatedly mentioned.

Response: The repeated statement has been reshuffled (Line number: 60-61) 

Lines 92 -103: Plagiarism noted from another article from the same author

Response: The sentences has been modified or reshuffled (Line number: 92-106). 

Please explain the number of samples collected per site. Were samples collected on a single visit for each of the site? Response: The ten samples of each three types (CDW, CBS and KS) were collected from each site in a single visit. The sampling frequency and time has been added in the “study design and sample collection” section as suggested by the reviewer (Line number: 134-135). 

Figure 1: Title should explain the inset and the pointers marked in the figure

Response: The title of figure-1 is modified in line with reviewer comments. The point is marked with blue color (Line number: 141). 

Plagiarism- Lines 149-150, 156 – 160

Response: The sentences has been reshuffled/modified and or replaced (Line number: 149-157 and 159-165). 

Table 1: Please include the target gene names

Response: The target gene name is included in the table 1 (212-213 line number) 

Please explain MAR

Response: MAR is explained in the manuscript (Line number: 194-199). 

It would be better to represent Figure 2A-C as a table with the percentage of resistance

Response: Figure 2A-C has modified into table (Table 3, 4 and 5); Similarly, figure 3 modified to table (Table 6, 7 and 8).

sul3 gene was detected in both KS and CBS. Please correct line 279

Response: It is corrected in the text (Line number: 304-306). 

Please explain “There was a strong correlation exist among the virulence genes in CDW, CBS, and KS (p < 0.005)” (line 293). It would be better to give figure 4 as a table as it is difficult to understand the results and, to identify the correlation between samples.

Response: The analysis showed significantly higher associations among the virulence genes in CDW, CBS and KS (p < 0.005). The detail statistical analysis is given in Supplementary file 2. Figure 4 has converted to Table 9 as suggested by the reviewer (Line Number: 326-328).

Please rewrite the discussion section to contain only relevant data pertaining to the results

Response: The discussion section has been modified as suggested by the reviewer. Reflected in the discussion section (Line: 332-528). 

416- 419-Plagiarism noted

Response: The statements are modified and reshuffled (Line number: 438-444).

Please add reference for the statement Line 475

Response: The reference (Chmielewski and Frank, 2003) is included in the text as suggested (Line number: 643-645). 

Line 344 – “Furthermore, China and some European countries detected S. Enteritidis and S. Typhimurium as the most prevalent serotypes” Did not explain from where they detected the serotypes The sources are catering point and meat of pork, chicken and duck.

Response: The information is included in the text (Line number: 379-381).

---

## [Editor Report · Decision Letter 1]

14 Jan 2022

Virulence and antimicrobial resistance profile of non-typhoidal Salmonella enterica serovars recovered from poultry processing environments at wet markets in Dhaka, Bangladesh

PONE-D-21-19124R1

Dear Dr. Samad,

We’re pleased to inform you that your manuscript has been judged scientifically suitable for publication and will be formally accepted for publication once it meets all outstanding technical requirements.

Kind regards,

Kumar Venkitanarayanan, DVM, Ph.D.

Academic Editor

PLOS ONE

---

## [Editor Report · Acceptance letter]

27 Jan 2022

PONE-D-21-19124R1 

Virulence and antimicrobial resistance profile of non-typhoidal *Salmonella enterica* serovars recovered from poultry processing environments at wet markets in Dhaka, Bangladesh 

Dear Dr. Samad:

I'm pleased to inform you that your manuscript has been deemed suitable for publication in PLOS ONE. Congratulations! Your manuscript is now with our production department. 

Kind regards, 

on behalf of

Dr. Kumar Venkitanarayanan 

Academic Editor

PLOS ONE